

# Officia
## Zarządzanie firmą usługową



**Autorzy**: Norbert Kinas · Dominik Patrzek · Miłosz Respondek · Krzysztof Śródka

**Opiekun:** Maja Kędras

**Streszczenie**

Projekt **Officia** koncentruje się na rozwiązaniu kluczowych problemów operacyjnych małych i średnich firm usługowych, takich jak zarządzanie danymi, niska efektywność oraz brak elastyczności systemów. Główne cele projektu obejmują automatyzację procesów, poprawę efektywności operacyjnej oraz skalowalność aplikacji. Wyróżniając się modułowością i integracją funkcji zarządzania klientami, pracownikami i zleceniami, aplikacja umożliwia dostosowanie do specyficznych potrzeb użytkowników.

W ramach projektu zrealizowano funkcjonalności, takie jak m.in. zarządzanie usługami, zleceniami, pracownikami, klientami, personalizacja interfejsu i konfigurowalność usług. Efekty obejmują m.in. redukcję czasu obsługi zleceń oraz zapewnienie wysokiej wydajności systemu.

Realizacja projektu potwierdziła jego potencjał w usprawnianiu procesów biznesowych, jednocześnie wskazując możliwości przyszłego rozwoju. System **Officia** stanowi efektywne narzędzie dla firm usługowych, zwiększając ich konkurencyjność na rynku.

# 1 TREŚĆ WŁAŚCIWA

## 1.1 Wstęp

**Officia** rozwiązuje kilka kluczowych problemów które są obecne w powszechnie stosowanych rozwiązaniach w małych i średnich firmach usługowych takich jak:

- problemy ze sprawnym zarządzaniem danymi,

- niska efektywność operacyjna,

- brak elastyczności.

Główne **cele biznesowe** projektu to:

- **Oszczędności czasowe:** poprzez automatyzację zadań związanych z fakturowaniem oraz zarządzaniem danymi o pracownikach, klientach i zleceniach.

- **Poprawa efektywności:** możliwość łatwego śledzenia zleceń i ich postępów przez pracowników, co przyczyni się do ich szybszej realizacji i optymalizacji procesów.

- **Skalowalność i konfigurowalność:** możliwość dostosowania aplikacji do specyficznych potrzeb organizacji zwiększy jej użyteczność i efektywność, co przyczyni się do lepszego zaspokojenia potrzeb klientów oraz zwiększy grono potencjalnych odbiorców.

- **Elastyczne dostosowywanie się do rynku:** aplikacja zostanie zaprojektowana w sposób umożliwiający dodawanie nowych funkcjonalności w przyszłości.

Rozwiązanie **Officia** przynosi znaczące korzyści, zarówno w postaci zwiększonej efektywności operacyjnej, jak i satysfakcji pracowników, co w dłuższej perspektywie zapewni wzrost zysków oraz konkurencyjności firm usługowych.

## 1.2 Prace związane z tematem

**Istniejące rozwiązania**:
Na rynku istnieje wiele systemów do zarządzania procesami w firmach, takich jak:

- HubSpot CRM, Zoho CRM, Pipedrive – koncentrują się na usprawnieniu procesów związanych z pozyskiwaniem i utrzymaniem klientów, ale nie obejmują innych obszarów działalności firmy, takich jak zarządzanie finansami, pracownikami, czy zadaniami.

- Trello, Asana, Monday.com – platformy do zarządzania projektami i zadaniami, jednak niezintegrowane z CRM i zarządzaniem pracownikami.

- ERP-y dedykowane dużym firmom (np. SAP, CEE, Microsoft Dynamics) – są często zbyt skomplikowane i kosztowne dla małych i średnich przedsiębiorstw. [10]

**Wyróżniki projektu *Officia***:

- **Zintegrowane podejście:** W odróżnieniu od rozproszonych narzędzi, nasza aplikacja łączy w jednym systemie funkcje zarządzania klientami, pracownikami i zleceniami.

- **Skalowalność i modułowość:** Aplikacja została zaprojektowana z myślą o łatwej rozbudowie, co umożliwia wprowadzanie nowych funkcji, takich jak zarządzanie magazynem czy analizy sprzedaży.

- **Dostępność:** Nasze rozwiązanie będzie dostępne w modelu atrakcyjnym dla małych i średnich firm.

**Wybrane rozwiązania technologiczne**:

- Backend:

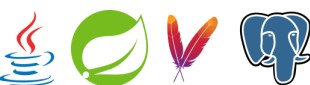

- Frontend:

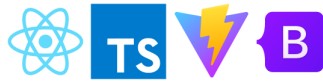

- Narzędzia pomocnicze:

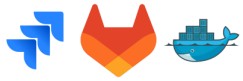

**Ograniczenia czasowe**:
Na opracowanie całego projektu wyznaczony został okres 10 tygodni.

**Zasoby**:
Zasoby ludzkie wykorzystane w ramach realizacji projektu obejmowały czterech studentów. Ponadto, w trakcie realizacji projektu regularnie odbywały się spotkania konsultacyjne z prowadzącymi, którzy służyli pomocą i doradzali w zakresie technicznym oraz merytorycznym.

**Napotkane problemy**:
Podczas realizacji projektu napotkano kilka istotnych wyzwań, wynikających zarówno z ograniczeń technicznych, jak i organizacyjnych:

- **Darmowe wersje narzędzi:** W przypadku korzystania z darmowej wersji GitLab, szybko wyczerpał się dostępny czas dla runnerów wbudowanych w platformę. Aby kontynuować pracę nad pipeline'ami CI/CD, potrzebne było skonfigurowanie własnego runnera na lokalnym serwerze.

- **Sprzęt:** Praca nad projektem odbywała na prywatnym sprzęcie, który w wielu przypadkach nie był najnowszy ani wystarczająco wydajny, co wpływało na czas budowania projektu oraz uruchamiania kontenerów Docker.

- **Ograniczenia czasowe:** Realizacja projektu odbywała się w stosunkowo krótkim czasie, co wymagało dokładnego priorytetyzowania zadań i efektywnego zarządzania czasem zespołu wykorzystując metodykę zwinną. [4]

- **Zrozumienie potrzeb klientów końcowych:** Kluczowym wyzwaniem było określenie realnych potrzeb klientów końcowych oraz dostosowanie funkcji aplikacji do ich oczekiwań. Proces ten wymagał prowadzenia analiz rynkowych i konsultacji, co było czasochłonne, ale niezbędne dla sukcesu projektu.

## 1.3 Wyniki

Głównym celem projektu było opracowanie aplikacji webowej umożliwiającej efektywne zarządzanie procesami biznesowymi w firmie. W trakcie realizacji osiągnęliśmy założone cele zarówno techniczne, jak i biznesowe.

### 1.3.1 Zaimplementowane funkcjonalności

- **Zarządzanie zleceniami:** Tworzenie zleceń od klientów zawierających zadania do wykonania dla pracowników. Zadania są tworzone na podstawie szablonów usług oferowanych przez firmę. Pracownik w trakcie wykonywania czynności może zmieniać aktualizować zadania o różne detale, takie jak użyte produkty, czy aktualny status.

- **Dokumentowanie działań:** Po wykonaniu wszystkich zadań system umożliwia wygenerowanie faktury, która zawiera szczegółowe informacje o wykonanych usługach. Proces ten jest zautomatyzowany, co zapewnia szybkie i bezbłędne wystawienie faktury, eliminując błędy ludzkie i konieczność ręcznego obiegu dokumentów.

- **Personalizacja:** Użytkownik posiada wiele opcji w aplikacji do wyboru. Może zmieniać język, motywy, używane waluty.

- **Konfigurowalność:** Pracownik definiując usługę, produkt uzupełnia postawowe pola, może również zdefiniować pola niestandardowe (custom fields), które dotyczą stricte danego obiektu i będą potrzebne pracownikowi do wykonania zadania.

- **Zarządzanie dostępem:** W systemie istnieje możliwość zarządzania uprawnieniami użytkowników. Administrator może przydzielać różne poziomy dostępu w zależności od roli pracownika w firmie, co zapewnia bezpieczeństwo danych i kontrolę nad tym, kto ma dostęp do jakich funkcji.

### 1.3.2 Osiągniete cele biznesowe i techniczne

Projekt pozwolił na usprawnienie procesów biznesowych oraz spełnienie wymagań technicznych:

- Oszczędność średnio 15 minut dziennie dla każdego pracownika, który nie musi już ręcznie zapisywać danych manualnie, co pozwala przeznaczać czas na inne zadania.

- Poprawa efektywności dzięki łatwemu śledzeniu zleceń i ich postępów przez pracowników.

- System jest w pełni skalowalny dzięki zastosowaniu architektury mikroserwisowej. [12]

# 2 WNIOSKI

## 2.1 Wnioski

Projekt **Officia** zakończył się sukcesem, osiągając kluczowe cele zarówno w zakresie automatyzacji procesów, jak i poprawy efektywności operacyjnej dla małych i średnich firm usługowych. Najważniejszym sukcesem projektu było opracowanie skalowalnej i elastycznej aplikacji webowej, która integruje zarządzanie klientami, pracownikami oraz usługami w jednym systemie. To rozwiązanie stanowi realną wartość dla biznesów, umożliwiając im usprawnienie procesów, co w rezultacie przekłada się na oszczędności czasowe i zwiększoną konkurencyjność na rynku.

Dzięki zastosowaniu systemu ról i różnych poziomów dostępu, **Officia** idealnie sprawdzi się w firmach, które zatrudniają pracowników z różnymi uprawnieniami i wymagają precyzyjnego zarządzania dostępem do funkcji aplikacji. Dodatkowo, elastyczność systemu oraz jego konfigurowalność sprawiają, że może on być wykorzystywany przez szeroki wachlarz branż, które mają specyficzne potrzeby w zakresie zarządzania danymi, pracownikami i zleceniami.

## 2.2 Kierunki rozwoju

Projekt posiada wiele perspektyw rozbudowy, które można zaimplementować w przyszłości.

- **Rozbudowa modułu zarządzania magazynem:** Wprowadzenie możliwości zarządzania wieloma magazynami jednocześnie, śledzenia stanów magazynowych w czasie rzeczywistym oraz automatycznego generowania raportów o stanach magazynowych.

- **Integracja z systemami zewnętrznymi:** Umożliwienie integracji z systemami API firm logistycznych w celu automatyzacji wysyłek, generowania etykiet oraz śledzenia przesyłek. Integracja z bazą CE-IDG - usprawnienie generowania faktur.

- **Obsługa kodów kreskowych i QR:** Implementacja funkcjonalności skanowania kodów kreskowych oraz kodów QR, co pozwoli na szybkie i intuicyjne dodawanie produktów do systemu lub aktualizację stanów magazynowych.

- **System promocji i rabatów:** Stworzenie zaawansowanego modułu umożliwiającego konfigurację promocji, rabatów sezonowych oraz pakietów promocyjnych. Moduł analityczny pozwoli ocenić efektywność prowadzonych działań marketingowych.

- **Integracja z drukarką fiskalną:** Zapewnienie obsługi drukarek fiskalnych zgodnych z wymogami prawnymi, umożliwiające drukowanie wygenerowanych paragonów oraz faktur.

- **Panel klienta:** Umożliwienie użytkownikom końcowym śledzenia statusów swoich zamówień w czasie rzeczywistym, składania zamówień przez internet oraz komunikacji z obsługą systemu.

- **Integracja z narzędziami komunikacyjnymi:** Wdrożenie możliwości integracji z platformami takimi jak Slack, Microsoft Teams czy WhatsApp, co ułatwi komunikację wewnątrz zespołów oraz kontakt z klientami.

- **Obsługa płatności online:** Połączenie systemu z popularnymi bramkami płatniczymi, takimi jak PayPal, Blik czy PayU, w celu umożliwienia szybkich i bezpiecznych płatności.

Powyższe kierunki rozwoju zapewnią elastyczność oraz lepsze dopasowanie systemu do wymagań użytkowników końcowych, co w przyszłości może przyczynić się do zwiększenia konkurencyjności aplikacji na rynku.

## 2.3    Bibliografia

- Dokumentacja Apache Maven [1]

- Unified Modeling Language [2]

- React Documentation [3]

- Method for Adaptation and Implementation of Agile Project Management Methodology [4]

- What I wish someone told me about Postgres [5]

- The Ultimate Guide to Bootstrap CSS [6]

- How-To Dockerize a Web Application [7]

- Agile UX Design for a Quality User Experience [8]

- Spring Framework Documentation [9].

- Koszt ERP [10]

- gRPC Fundamental and Concept [11]

- Mikroserwisy [12]

## LITERATURA

[1] *Dokumentacja Apache Maven.* [Link].

[2] *Unified Modeling Language.* [Link].

[3] Jason Bonta Joe Savona Josh Story Lauren Tan Luna Ruan Mofei Zhang Rick Hanlon Samuel Susla Sathya Gunasekaran Sebastian Markbåge Sebastian Silbermann Seth Webster Sophie Alpert Tianyu Yao Yuzhi Zheng Andrew Clark, Dan Abramov. *React Documentation.* [Link].

[4] Solvita Berzisa Arturs Rasnacis. *Method for Adaptation and Implementation of Agile Project Management Methodology.* [Link].

[5] Hazel Bachrach. *What I wish someone told me about Postgres.* [Link].

[6] Anna Fitzgerald. *The Ultimate Guide to Bootstrap CSS.* [Link].

[7] Sasi Kumar. *How-To Dockerize a Web Application.* [Link].

[8] Pardha S. Pyla Rex Hartson. *Agile UX Design for a Quality User Experience.* [Link].

[9] Keith Donald Colin Sampaleanu Rob Harrop Thomas Risberg Alef Arendsen Darren Davison Dmitriy Kopylenko Mark Pollack Thierry Templier Erwin Vervaet Portia Tung Ben Hale Adrian Colyer John Lewis Costin Leau Mark Fisher Sam Brannen Ramnivas Laddad Arjen Poutsma Chris Beams Tareq Abedrabbo Andy Clement Dave Syer Oliver Gierke Rossen Stoyanchev Phillip Webb Rob Winch Brian Clozel Stephane Nicoll Sebastien Deleuze Jay Bryant Mark Paluch Rod Johnson, Juergen Hoeller. *Spring Framework Documentation.* [Link].

[10] Kevin Cyriac Tom. *Your 2024 ERP Price Guide – What Your System Will Cost You.* [Link].

[11] Pramono Winata. *gRPC Fundamental and Concept.* [Link].

[12] Jarosław Żeliński. *Mikroserwisy.* [Link].
