# OpenReview forum: "Officia - Zarządzanie firmą usługową"
_pwr.edu.pl/Wrocław_University_of_Science_and_Technology/2024/ZPI_Day — Wrocław University of Science and Technology 2024 ZPI Day Submission_

### Official Review · Reviewer_Qazk · 2024-12-04
**Officia**

**Confidence:** 5
**Significance Of Results:** 3
**Overall Quality:** 3

**Compliance With Template:**

3: Average Quality – The article includes most of the required sections, but some may be incomplete, written in a general or unclear manner. The content is correct but requires further refinement.

**Description Of Results:**

3: Average Quality – The results are described with moderate detail. Some examples or evaluation elements are present but insufficiently developed or incomplete.

**Feedback On Consistency:**

Zdublowana sekcja z literaturą i bibliografią. Referencje niepełne (w zamierzeniu studentów chyba były linki). Studenci starali się wypełnić wszystkie zalecenia szablonu, ale poza opisami, nie ma żadnego dowodu, że produkt powstał, np. zrzutów ekranu. Porównanie z innymi produktami dość ogólne, tak jak i cały dokument. Innowacyjność produktu niska.

**Potential For Development:**

Tak. Wskazano wiele kierunków rozwoju produktu.

**Project Nature Evaluation:**

Opis techniczny mało czytelny - mamy same ikonki technologii, bez legendy czy podpisów. Wiadomo tylko, że użyto architektury mikroserwisowej, ale nie ma żadnych szczegółów na temat front-endu oraz bazy danych.

**Technical Language Precision:**

4: High Quality – The language is appropriate for a technical report. Terminology is used correctly, and statements are precise, with only minor shortcomings that do not affect the overall clarity.

---

### Official Review · Reviewer_k9qQ · 2024-12-06
**An user-centric project, well-suited for future development**

**Confidence:** 4
**Significance Of Results:** 5
**Overall Quality:** 5

**Compliance With Template:**

5: Very High Quality – The article contains all the required sections, which are written in a very detailed, clear, and error-free manner. The structure is professional and meets expectations, and the content adheres to the highest substantive and formal standards.

**Description Of Results:**

5: Very High Quality – The results are described in detail, clearly and comprehensively, supported by thorough evaluation, analysis, and convincing usage examples. The description meets the highest substantive standards.

**Feedback On Consistency:**

The problem analysis, presentation of results, and conclusions are mostly consistent and logical. The article provides a clear link between the problem, the solutions implemented, and the results achieved. Additionally, the description of the problems encountered along the way is valuable and helps to understand the challenges faced during the project.

**Potential For Development:**

The article shows clear possibilities for further work and practical use of the results. The future directions mentioned, such as connecting with external systems, adding new features, and expanding to larger companies, indicate that the project has great potential for growth.

**Project Nature Evaluation:**

The project clearly exhibits characteristics of an engineering work. It demonstrates practical application and addresses real-world problems. The selected technologies and solutions are justified in the context of identified needs and problems. However, additional details regarding the methods used to verify the success indicators of the project would be beneficial.

**Technical Language Precision:**

5: Very High Quality – The language is entirely appropriate for a technical report. All terms are used correctly and precisely, and the style is professional, clear, and coherent, without any errors or ambiguities.

---

### Official Review · Reviewer_5pFe · 2024-12-07
**Małe rozwiązanie dla małych firm, duże obietnice.**

**Confidence:** 5
**Significance Of Results:** 3
**Overall Quality:** 3

**Compliance With Template:**

3: Average Quality – The article includes most of the required sections, but some may be incomplete, written in a general or unclear manner. The content is correct but requires further refinement.

**Description Of Results:**

2: Low Quality – The results are described very superficially and in a general manner. Essential details, usage examples, or evaluations are missing.

**Feedback On Consistency:**

Opis projektu jest zgodny z szablonem, ale bardzo pobieżny. Autorzy konsekwentnie wspominają o oszczędnościach, integracji, automatyzacji, ale w samym dokumencie trudno było mi znaleźć konkrety uzasadniające te hasła.
W przeglądzie istniejących rozwiązań autorzy wymieniają systemy CRM i zarządzania projektami, natomiast w opisie wyników mówią o zarządzaniu zleceniami i dokumentowaniu zadań, czyli o zupełnie innych tematach. Zarządzanie klientami omówione w pracy to tylko niewielki wycinek funkcji oferowanych przez systemy CRM.
W pracy, nie wiem w jakim celu, umieszczono dwie właściwie identyczne sekcje: Bibliografię i Literaturę.

**Potential For Development:**

Artykuł zawiera opis możliwości rozwoju aplikacji w przyszłości. Oprócz przedstawionych opcji, warto byłoby rozważyć dodanie możliwości konfigurowania przepływu pracy. "Konfigurowalność" o której wspominając autorzy odnosi się tylko do możliwości dodawania pól niestandardowych do istniejących usług i obiektów (choć ze względu na ograniczony czas projektu jest to zrozumiałe).
Aplikacja ma potencjał do praktycznego zastosowania w przyszłości, zwłaszcza w dziale małych firm, potrzebujących szybkich i niewielkich systemów realizujących wybrane funkcje.

**Project Nature Evaluation:**

Opis pracy jest bardzo pobieżny i na jego podstawie trudno ocenić jego zaawansowanie techniczne. Autorzy wspominają o wykorzystaniu podejścia bazującego na mikrousługach, jednak nigdzie w zgłoszeniu nie opisano (przynajmniej ogólnie) w jaki sposób zostały one zaimplementowane.
Autorzy nie uzasadnili również czy w aplikacji, takiej jak ta, przeznaczonej dla małych firm, jest potrzebna ogromna skalowalność?

**Technical Language Precision:**

3: Average Quality – The language is mostly appropriate but may contain minor terminological or stylistic errors. Some statements might lack precision or require improvement for better readability.

---

### Decision · Program_Chairs · 2024-12-10

Accept (Poster)